# Weather-Based Statistical and Neural Network Tools for Forecasting Rice Yields in Major Growing Districts of Karnataka

Mathadadoddi Nanjundegowda Thimmegowda [1,\*], Melekote Hanumanthaiah Manjunatha [1], Lingaraj Huggi [1], Huchahanumegowdanapalya Sanjeevaiah Shivaramu [2], Dadireddihalli Venkatappa Soumya [1], Lingegowda Nagesha [1] and Hejjaji Sreekanthamurthy Padmashri [3]

1   AICRP on Agrometeorology, University of Agricultural Sciences, GKVK, Bengaluru 560065, India
2   College of Horticulture, University of Agricultural Sciences, Kolar 563103, India
3   Directorate of Research, university of Agricultural Sciences, Gkvk, Bengaluru 560065, India
\*   Correspondence: bng.aicrpam@gmail.com

**Abstract:** Two multivariate models were compared to assess their yield predictability based on long-term (1980–2021) rice yield and weather datasets over eleven districts of Karnataka. Simple multiple linear regression (SMLR) and artificial neural network models (ANN) were calibrated (1980–2019 data) and validated (2019–2020 data), and yields were forecasted (2021). An intercomparison of the models revealed better yield predictability with ANN, as the observed deviations were smaller (−37.1 to 21.3%, 4% mean deviation) compared to SMLR (−2.5 to 35.0%, 16% mean deviation). Further, district-wise yield forecasting using ANN indicated an underestimation of yield, with higher errors in Mysuru (−0.2%), Uttara Kannada (−1.5%), Hassan (−0.1%), Ballari (−1.5%), and Belagavi (−15.3%) and overestimations in the remaining districts (0.0 to 4.2%) in 2018. Likewise, in 2019 the yields were underestimated in Kodagu (−0.6%), Shivamogga (−0.1%), Davanagere (−0.7%), Hassan (−0.2%), Ballari (−5.1%), and Belagavi (−10.8%) and overestimated for the other five districts (0.0 to 4.8%). Such model yield underestimations are related to the farmers' yield improvement practices carried out under adverse weather conditions, which were not considered by the model while forecasting. As the deviations are in an acceptable range, they prove the better applicability of ANN for yield forecasting and crop management planning in addition to its use for regional agricultural policy making.

**Keywords:** statistical model; SMLR; ANN; rice yield; weather

## 1. Introduction

Rice (*Oryza sativa* L.) is the most important staple food crop of India, next to wheat; is used for food and animal fodder; and is cultivated in a 45.76 million ha area, with a production 124.3 million tons in the country. Karnataka is the major rice-growing state in India, contributing 3 percent of the country's rice area (1.397 million ha) and 3.45 percent (4.29 million tons) of production [1]. The crop is cultivated in a wide range of soils and rainfall and temperature situations. As a unique example, it is cultivated in areas where rainfall ranges from 600 to 3000 mm per annum [2]. The unique feature of rice culture in Karnataka is that either sowing or transplanting is seen in all seasons of the year, and the durations of cultivated rice varieties vary from 100 to 180 days, depending on the season and agroclimatic conditions. Despite its ability to adapt to a wide range of climatic conditions, the crop suffers from severe yield variability due to changes in weather factors [3]. Weather factors affect the crop both in direct and indirect ways: directly, as a source of water for crop growth and as an energy source for physiological aspects though light and temperature, and indirectly through the mineralization of nutrients, their movement to the plant root zone, etc. [4,5]. Further, these direct and indirect impacts of weather factors are the results of individual weather factors or the interactive effect of two or more weather factors on

the crop yield. Previously, a lot of work has been carried out to estimate the impacts of individual weather factors on crop yield [6–8], but few recent studies have shown the interactive effects of weather factors on crop performance [9,10]. Studies in this direction will help in understanding crop responses in terms of final yield and provide a forecast of the crop prior to harvest [11].

The estimation of the interactive effects of weather factors on crop yield are aided by advancements in crop yield forecasting techniques, which predict a crop yield based on the weather variables that prevailed during the crop growth period. Such forecasting techniques, commonly called 'crop models' are handy in crop planning, as they are developed based on multidisciplinary sources of information such as edaphic (land use, soil physical properties, soil pH, soil fertility, soil moisture, etc.) [12], meteorological (temperature, rainfall, relative humidity, etc.) [13], management (row spacing, seed quantity, the fertilizers and pesticides used, etc.), and crop factors (genotype x environment interaction) [14]. Such models are classified into three groups based on the input requirements [15]: empirical or statistical models, simulation models, and weather analysis models. All of these models rely upon two important aspects: the usage of the traditional approach of mathematical models and the application of artificial intelligence [16].

Previously, in many studies statistical methods such as multiple linear regressions (MLRs) have been employed to develop statistical crop yield prediction models [17–19]. These models should be used cautiously, as there is a chance of model overfitting because of the overdependence of the dependent factor (yield) on independent factors (weather variables), as the independent factors are known for multicollinearity [20]. To overcome the problem of model overfitting, methods such as stepwise multiple linear regression (SMLR), artificial neural network (ANN), least absolute shrinkage and selection operator (LASSO), or elastic net (ENET) have been adopted in many previous studies to increase the precision in yield forecasts [21]. The SMLR models only consider major factors responsible for yield formation. The technological advancements have also brought solutions for complex agricultural problems that linear systems are unable to resolve. One such advancement is the use of neural networks; they take into account the multidirectional interactions between independent variables to precisely simulate the dependent variable [22]. Previously, many attempts were made in this direction using statistical and simulation models [23–25]. Here, an effort was made to establish and compare statistical (SMLR) and neural network (ANN) models for yield estimation to arrive at a better model with the intention of aiding regional policy making.

## 2. Materials and Methods

### 2.1. Study Area

The major rice districts of Karnataka were selected for the study. The majority of the districts were judged based on the area and production of the crop; these eleven districts contribute to roughly 50 percent of the state's rice area and production. In total, eleven districts (Table 1, Figure 1a,b) were selected, considering their contributions to production over many years. As the crop was introduced to new districts in recent years, the fear of dataset unavailability prevented us from choosing those districts. Among the eleven districts, the highest area, production, and productivity of rice (178.5 thousand ha, 723.0 thousand t, and 4051.2 kg/ha, respectively) were observed in Ballari district, and lower levels were observed in Dakshina Kannada (DK) district (8.1 thousand ha area and 23.4 thousand t of production). Lower productivity of the crop was noticed in Uttara Kannada (UK) district (2149.3 kg/ha).

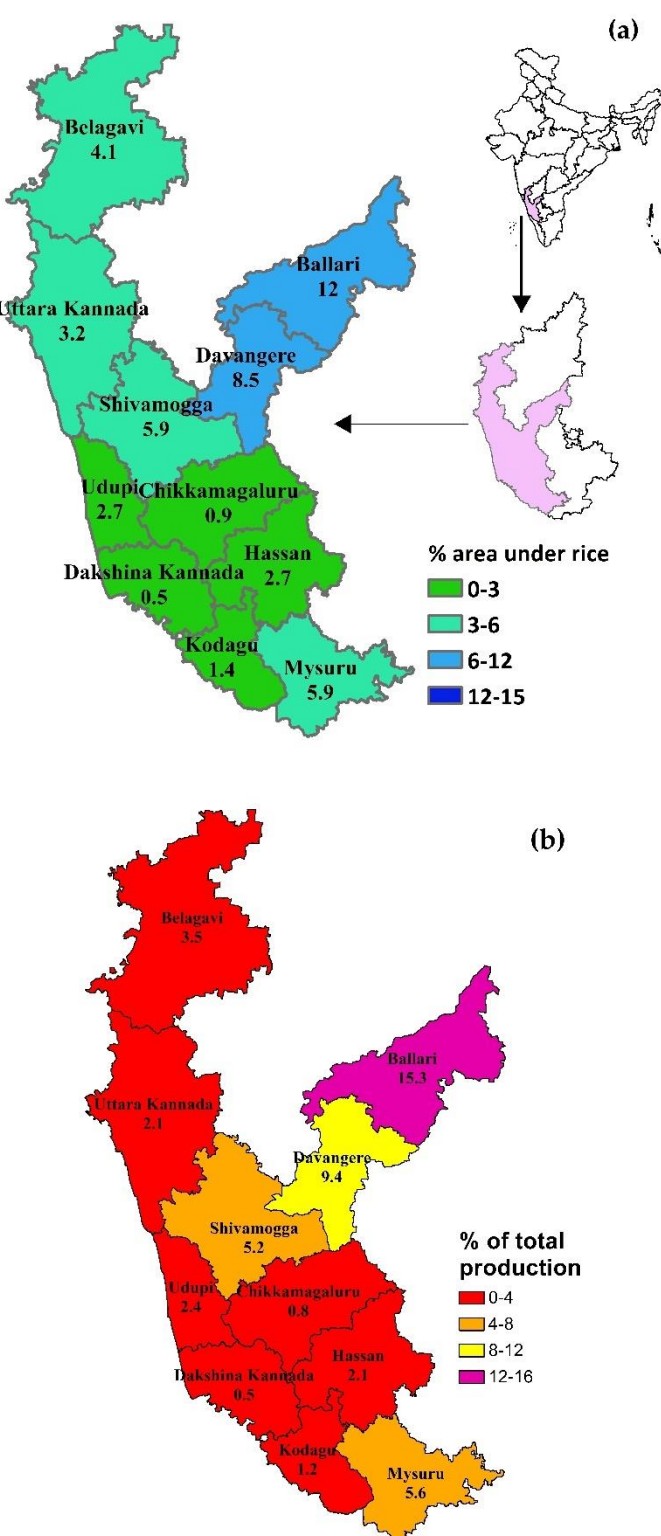

**Figure 1.** Percentage of rice area (**a**) and percent contribution to state's rice production (**b**) by each district of the study area (Source: Rice area and production (2021), Directorate of Economics and Statistics, GoK).

**Table 1.** Area, production, and productivity of rice (2020).

| Districts | Rice Area * | Production ** | Productivity (kg/ha) |
|---|---|---|---|
| Ballari | 178.5 | 723.0 | 4051.2 |
| Belagavi | 60.7 | 166.7 | 2744.3 |
| Chikkamagaluru | 14.0 | 37.5 | 2668.3 |
| Dakshina Kannada | 8.1 | 23.4 | 2883.4 |
| Davanagere | 126.1 | 444.2 | 3523.7 |
| Hassan | 39.6 | 99.1 | 2502.9 |
| Kodagu | 21.3 | 56.4 | 2643.1 |
| Mysuru | 87.1 | 263.1 | 3021.4 |
| Shivamogga | 88.2 | 245.1 | 2778.1 |
| Udupi | 40.2 | 115.3 | 2870.7 |
| Uttara Kannada | 47.0 | 101.0 | 2149.3 |
| Karnataka | 1484.0 | 4717.5 | 3178.0 |

* Area in thousand hectares. ** Production in thousand tons.Source: Rice area and production (2021), Directorate of Economics and Statistics, GoK.

### 2.2. Dataset

Long-term (42 years; 1980 to 2021, Table S1) datasets pertaining to the area, production, and productivity of rice in the state were collected from the Directorate of Economics and Statistics, Government of Karnataka. Daily weather data (maximum and minimum temperature, morning and evening relative humidity, and rainfall) pertaining to the study years were collected from the India Meteorological Department, Pune, for the districts under study.

### 2.3. Methodologies Used for Yield Forecast

Two approaches of crop yield forecast are in vogue recently [26]. One is the data-intensive, cumbersome simulation model. These have limited applicability since they cannot be applied to large spatiotemporal scales due to the unavailability of sufficient input data. Therefore, the other method, statistical models using crop yield and weather data by means of simple regression, can be broadly used as an alternative to process weather-based statistical models [27]. For successful weather-based forecasting, statistical models should first be calibrated and tested using historical datasets. A district-wise yield model for rice in Bihar was developed using meteorological data, and it showed that models were able to predict preharvest crop yield with good accuracy. Most of the statistical models use multiple linear regression (MLR) equations to develop statistical crop yield prediction models [28]. To overcome multi-colinearity between independent variables, feature selection (stepwise multiple linear regression (SMLR), least absolute shrinkage and selection operator (LASSO), or the elastic net (ENET) method) or feature extraction (principal component analysis) statistical techniques can be used. In a few studies, PCA has been used in conjunction with MLR. However, studies on the comparison of the performance of models with and without feature selection, feature extraction, and a combination of both the methods for forecasting crop yield are meagre. In this context, our study has found an opportunity to develop and select a statistical forecasting model using SMLR and ANN for major rice-growing districts of Karnataka, with the objectives to (i) develop district-wise crop yield prediction models using multivariate models and (ii) evaluate the predictive performance of the developed models. The methodology followed for both SMLR and ANN are summarized in the next section.

#### 2.3.1. Generation of Weather Indices

Weather indices were generated based on composite weather variable methods. Two types of weather variables were generated, i.e., unweighted and weighted weather variables. Unweighted weather indices are calculated using the sum of weekly weather variables experienced during a crop period, while the weighted indices are calculated using the sum product of this correlation coefficient and the value of the weekly weather variable.

Correlation coefficients between the yield and the weather variables experienced during the respective week were calculated. Similar weather index-based yield forecasting model approaches were used for rice, wheat, sugarcane, and potato in Uttar Pradesh, India [29]. The procedure for the computation of unweighted and weighted weather indices is summarized below. In total, 42 weather variables were generated to determine their effects on the yield of rice.

Unweighted weather indices:

$$Z_{ij} = \sum_{w=1}^{m} X_{iw} Z_{ii'j} = \sum_{w=1}^{m} X_{iw} X_{i'w}$$

Weighted weather indices:

$$Z_{ij} = \sum_{w=1}^{m} r_{iw}^{j} X_{iw} Z_{ii'j} = \sum_{w=1}^{m} r_{ii'm}^{j} X_{iw} X_{i'w}$$

where:

- $X_{iw}/X_{ii'}$—the value of the $i$th/$ih$th weather variable understudy in the $w$th week;
- $rjiw/rjii/e$—the correlation coefficient of the detrended yield with the $i$th weather variable/product of the $i$th and $ia$ th weather variables in the $w$th week;
- $m$—the week of the forecast.

### 2.3.2. Simple Multiple Linear Regression (SMLR)

ICAR, the Indian Agricultural Statistical Research Institute (IASRI), has developed models that express the effect of weather variables on the yields of the respective correlation coefficients between the yield and weather variables. Here, the yield is considered a dependent variable, and the weekly weather variables are considered independent variables. The weekly weather variables were generated using daily data by averaging the daily maximum temperature (daily TMAX) and minimum temperature (daily TMIN) and the morning relative humidity (daily RHI) and evening relative humidity (daily RHII) and summing up the rainfall (daily RF) and were used for further analysis (Figure 2) and for the generation of the weather indices indicated in Table 2.

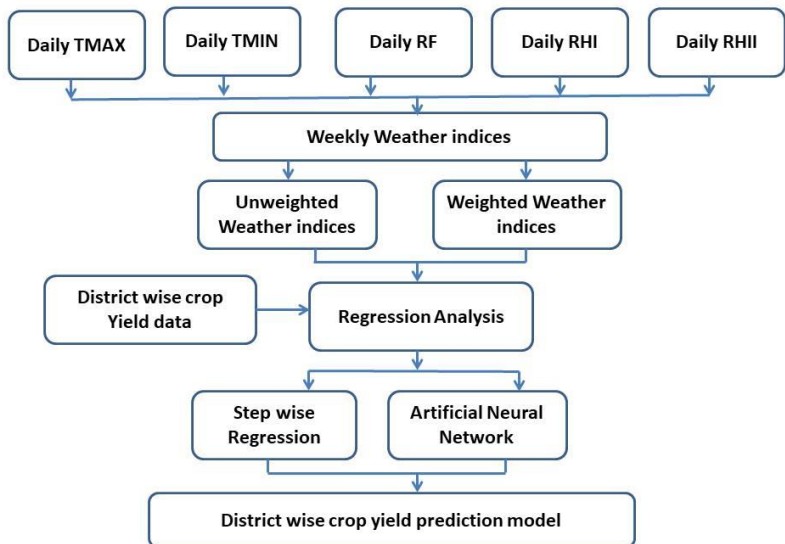

**Figure 2.** Flowchart representing the different stages of model calibration and validation.

Multiple linear regression (MLR) is the standard and simplest approach for the development of calibration models, but its application for datasets with independent variables

with large sample numbers is not always successful [30]. However, feature selection in the form of stepwise MLR (SMLR) gives good results over large datasets. A stepwise regression procedure was adopted for the selection of the best regression variable among many independent variables [31].

**Table 2.** Weather-derived indices used in models using composite weather variables.

| Weather Variables | Unweighted Weather Indices (0) | | | | | Weighted Weather Indices (1) | | | | |
|---|---|---|---|---|---|---|---|---|---|---|
| | TMAX (1) | TMIN (2) | RF (3) | RHI (4) | RHII (5) | Tmax (1) | Tmin (2) | RF (3) | RHI (4) | RHII (5) |
| Maximum temperature (1) | Z10 | | | | | Z11 | | | | |
| Minimum temperature (2) | Z120 | Z20 | | | | Z121 | Z21 | | | |
| Rainfall (3) | Z130 | Z230 | Z30 | | | Z131 | Z231 | Z31 | | |
| Morning relative humidity (4) | Z140 | Z240 | Z340 | Z40 | | Z141 | Z241 | Z341 | Z41 | |
| Evening relative humidity (5) | Z150 | Z250 | Z350 | Z450 | Z50 | Z151 | Z251 | Z351 | Z451 | Z51 |

The SMLR-based statistical inference relies upon the assumption that the sample mean is approximately normally distributed while testing the population mean. This necessitates checking for normality in the sample dataset [32]. Nearly 40 different normality tests have been developed and have proven their vitality in many statistical analyses by their different applicability values because the power would change owing to the sample size and the nature of the data [33]. Hence, one should be cautious when choosing an appropriate test of normality. In this study, the Shapiro–Wilk test [34] was used to test the normality of district-wise yield data.

### 2.3.3. Artificial Neural Networks (ANN)

Attaining the maximum crop yield at the minimum cost is one of the aims of agricultural production. Hence, the early detection and management of problems associated with crop yield indicators can help to increase yields. Recently, the application of artificial intelligence (AI), such as artificial neural networks (ANNs) (Figure 3), fuzzy systems, and genetic algorithms, has been shown to be more efficient in solving these problems. Using these processes can make models easier to use and more accurate when working with complex natural systems with many inputs. In the present study, we used three layers, namely input, hidden, and output feed-forward artificial neural network. Each layer had neurons or nodes interconnected with each other. The number of nodes in the input and output layers was fixed by the dataset used. There was a need to take care when choosing the optimal number of hidden layers while implementing the ANN for yield forecasting by using the 'train' function of the 'caret' package using the method '*nnet*' with 10-fold cross-validation in R software [35]. Here, the analysis was carried out by selecting 80 percent of the data for calibration (training) purposes and the remaining dataset for validation (testing). In the present study, 32 weather indices were used as inputs. Yield was the dependent variable, and the rest were independent variables.

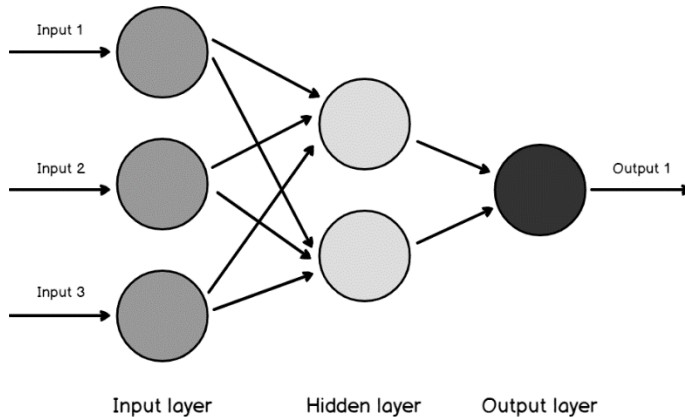

**Figure 3.** Diagrammatic representation of ANN.

*2.4. Tests of Model Performance*

Model performance was tested using different statistical model performance evaluation measures. The use of more than one measure helped us to evaluate a single model's performance and compare multiple models. In this study, the $R^2$, root-mean-square error (RMSE), normalized root-mean-square error (nRMSE), and modeling efficiency (EF) were calculated using the formulae:

$$R^2 = \left( \frac{\frac{1}{n} \sum_{i=1}^{n} \left( M_i - \overline{M} \right) \left( O_i - \overline{O} \right)}{\sigma_M \sigma_O} \right)^2$$

$R^2$ is the proportion of variation in the outcome that is explained by the predictor variables. In multiple regression models, it corresponds to the squared correlation between the observed outcome values and the values predicted by the model. The higher the R-squared (~1), the better is the model prediction.

$$RMSE = \sqrt{\frac{1}{n} \sum_{i=1}^{n} (O_i - M_i)^2}$$

This measures the average magnitude of the errors and is concerned with deviations from the actual value. An RMSE value of zero indicates that the model has a perfect fit. The lower the RMSE, the better the model and its predictions.

$$nRMSE = \sqrt{\frac{1}{n} \sum_{i=1}^{n} (O_i - M_i)^2} \times \frac{100}{\overline{O}}$$

$$EF = 1 - \frac{\sum_{i=1}^{n} (O_i - M_i)^2}{\sum_{i=1}^{n} \left( O_i - \overline{O} \right)^2}$$

$M_i$: model output; $\overline{M}$ and $\sigma_M$: mean and standard deviation of model output, respectively; $O_i$: observations; $\overline{O}$ and $\sigma_o$: mean and standard deviation of observations, respectively.

The normalized root-mean-square error expresses the spread around the measurements and is used for the classification of model performance into distinct groups (excellent, good, fair, and poor when the values are in the ranges of <10%, 10–20%, 20–30%, and >30%, respectively) [36], while the modeling efficiency indicates whether the model describes the data better than simply using the average of the predictions. Optimal values are the ones that are close to 1.

**3. Results**

*3.1. Observed Variability in Rainfall in the Study Districts*

Rice seeding in Karnataka starts in June (24th SMW) and mostly ends in September (39th SMW). The spatial and temporal (weekly) rainfall distribution during the study period over the region was calculated and is depicted in Figure 4a,b. The average rainfall of all the districts indicated the maximum rainfall in the 27th SMW (2–8th July), and the rainfall declined from the 28th SMW (16–22nd July). Most of the districts followed the same trend, except Dakshina Kannada, where the rainfall was comparatively higher throughout the crop growth period. Spatially, the three coastal districts, Udupi, Dakshina Kannada, and Uttara Kannada, received more rainfall during the crop growth period (194.4, 164.0, and 118 mm, respectively) and annually. These were followed by Malnad districts Shivamogga, Chikkamagaluru, and Kodagu (104.3, 80.6, and 102.8 mm, respectively), and the remaining four interior districts, Hassan, Mysuru, Davanagere, and Ballari (41.0, 41.9, 21.7, and 25.3 mm, respectively) received lower rainfall during the crop growth period. Many studies have indicated the impact of such an unequal spatial distribution of rainfall [37–40] on the yield variability of rainfed crops. However, crops such as rice are capable of producing yields under irrigated situations, and rainfall distribution has a meagre impact, although in

Karnataka only a few rice-growing districts are under irrigation (Ballari and Mandya), and in the remaining districts rice is grown in rainfed conditions. Hence, in the present study, only the *kharif* datasets on area, production, and productivity were collected and used.

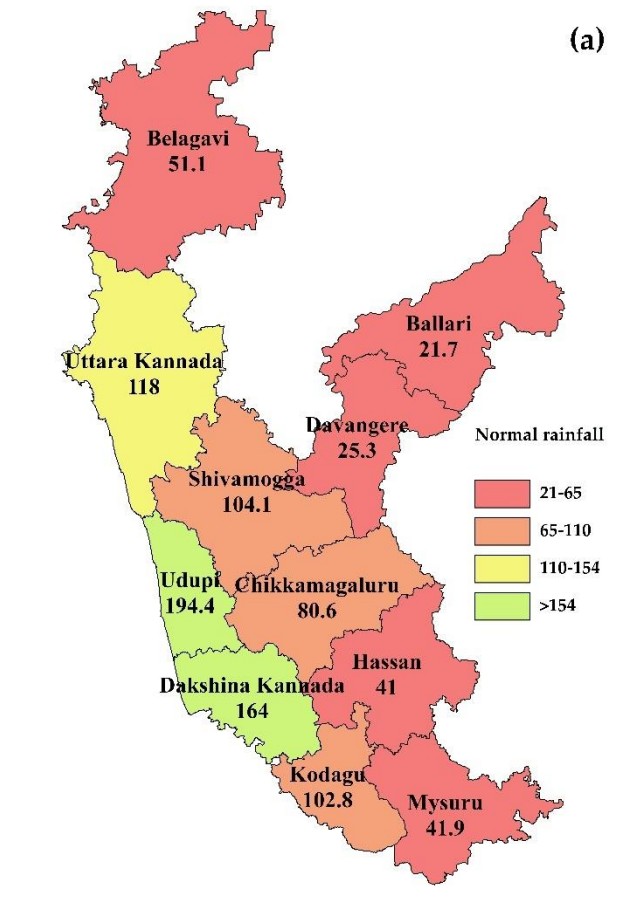

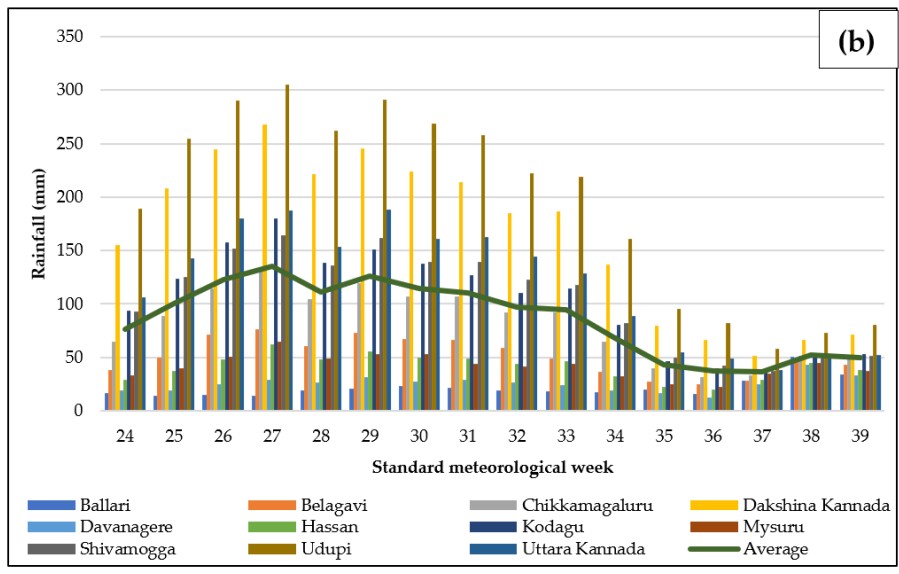

**Figure 4.** Spatial (**a**) and temporal (**b**) variability in rainfall in rice-growing districts during rice growing period, i.e., 24th to 39th SMW.

### 3.2. Description of Rice Yield Variability in the Study Districts

The summary statistics of yield data pertaining to the eleven rice-growing districts of Karnataka over the years 1980–2019 are presented in Table 3. The maximum yield from the collected data was observed in Davanagere district (3541.17 kg ha$^{-1}$), and the minimum yield was observed in Belagavi district (1950.63 kg ha$^{-1}$). The standard deviation of the yields across the districts varied between 244.57 and 643.37 kg ha$^{-1}$. Among the districts, higher yield variability (CV) was observed in Ballari and Belagavi (18.4 and 31.7%, respectively). Such variability in the *Kharif* rice yield in these districts was mainly attributed to the characteristic high temperatures in these districts, causing crop failures under minor changes in intra-annual rainfall distribution. Further, for the sake of fitting SMLR, the normality of the yield data was tested using normal Q–Q plots (Figure 5) and Shapiro–Wilk tests. The yield data were found to be normally distributed, as indicated by Shapiro–Wilk test ($p$ value > 0.05), for all districts except Mysuru ($p$ value = 0.003). The normal Q–Q plots also confirmed normality, as the quintiles almost formed a diagonal line, thereby fulfilling the basic assumption of the parametric models (MLR, LASSO, and ENET).

**Table 3.** Statistics of rice yield variability in eleven study districts.

| District | Mean | Maximum | Minimum | Std | CV (%) | Shapiro–Wilk Test | |
|---|---|---|---|---|---|---|---|
| | | | | | | Statistic | $p$ Value |
| Ballari | 3485 | 4571 | 2406 | 643.3 | 18.4 | 1.78 | 0.162 |
| Belagavi | 1951 | 3096 | 589 | 618.9 | 31.7 | 1.45 | 0.269 |
| Chikkamagaluru | 2547 | 3023 | 2062 | 249.5 | 9.8 | 1.36 | 0.266 |
| Dakshina Kannada | 2529 | 3375 | 2061 | 328.4 | 12.9 | 2.3 | 0.091 |
| Davanagere | 3541 | 4135 | 3114 | 244.5 | 6.9 | 0.53 | 0.774 |
| Hassan | 2653 | 3417 | 1602 | 444.3 | 16.7 | 0.93 | 0.510 |
| Kodagu | 2656 | 3126 | 2064 | 245.5 | 9.2 | 2.02 | 0.117 |
| Mysuru | 3147 | 3669 | 1968 | 305.3 | 9.7 | 38.87 | 0.003 |
| Shivamogga | 2710 | 3539 | 2024 | 346.8 | 12.8 | 0.71 | 0.056 |
| Udupi | 2674 | 3250 | 1904 | 311.1 | 11.6 | 1.58 | 0.505 |
| Uttara Kannada | 2055 | 2758 | 1462 | 288.3 | 14.0 | 0.26 | 0.433 |

### 3.3. Rice Yield Forecasting Models

3.3.1. Stepwise Multiple Linear Regression Model

The *Kharif* rice yields were forecasted during 2021 at the F3 (preharvest) stage using SMLR with SPSS statistical software for eleven districts (Belgavi, Dakshina Kannada, Davanagere, Hassan, Chikkamaglur, Udupi, Shivamogga, Ballari, Mysore, Kodagu, and Uttara Kannada). Its regression equation, the weather variables influencing the equation, and the standard error (SE) of the estimated values resulting from different weather variables are presented in Table 4. Here, a lower SE (94.99) was observed in Udupi district, and a higher SE (465.48) was seen in Belagavi district. The method of yield forecasting at various crop growth stages had a variable capability of yield estimation. The forecasts given at the midcrop (F2) and preharvest (F3) stages had better similarity to the observed yield compared to that forecasted at the vegetative stage, i.e., F1 [41]. Previously, such models have proven their worth in forecasting the yields of many crops, such as sugarcane and potato [42].

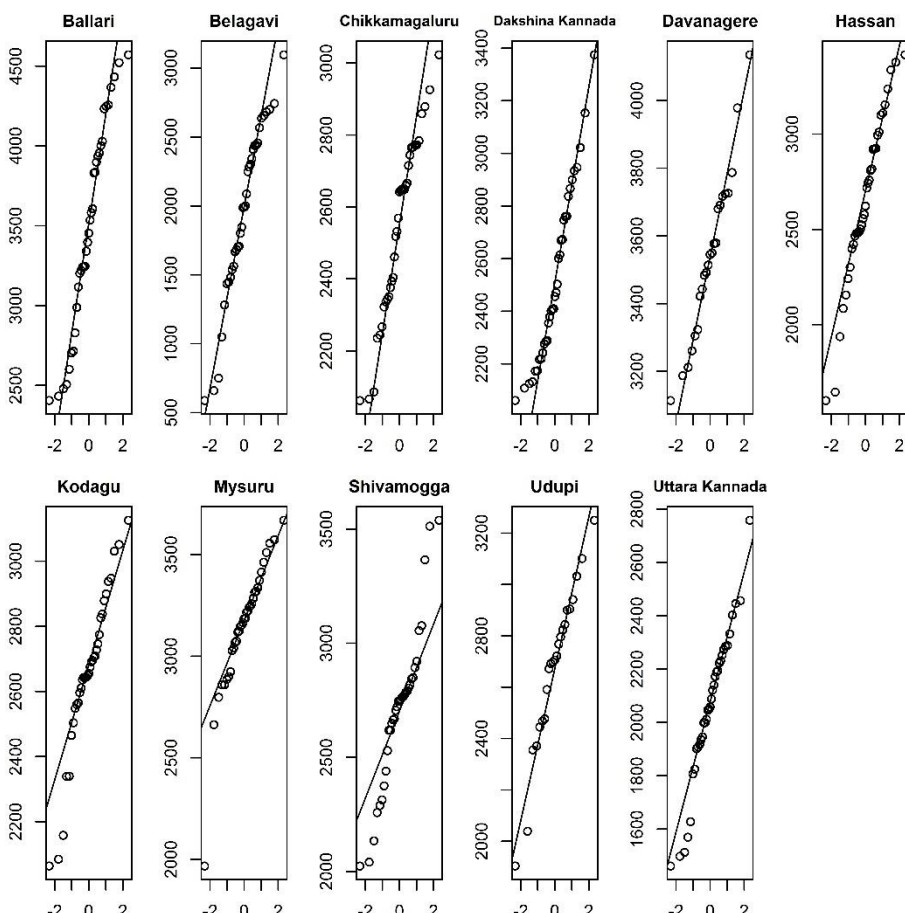

**Figure 5.** Normal Q–Q plot for *kharif* rice yields in 11 study districts of Karnataka.

**Table 4.** Rice yield prediction equations using stepwise multiple linear regression for different districts of Karnataka during 2021 at preharvest (F3) stage.

| District | Regression Equation | Weather Variables in the Equation | F | Std. Error |
|---|---|---|---|---|
| Ballari | Y = −13.44 + 35.87 *Time + 8.61 *Z10 + 0.04 *Z341 | Time, Tmax, Rf *Rh1 | 118.54 | 201.3 |
| Belagavi | Y = −145.69 + 0.25 *Z231 + 0.60 *Z251 | Tmin *Rf, Tmin *Rh2 | 19.39 | 465.48 |
| Chikkamagaluru | Y = −74.7731 + 22.23 *Time + 10.63 *Z10 + 0.26 *Z131 | Time, Tmax, Tmax *Rf | 50.52 | 248.45 |
| Dakshina Kannada | Y = 25.81 + 42.80 *Time-4.93 *Z51*0.40 *Z121 | Time, Rh2, Tmax *Tmin | 90.21 | 181.93 |
| Davanagere | Y = −61.0 + 30.14 *Time + 13.84 * Z10 + 0.16 *Z131 | Time, Tmax, Tmax *Rf | 115.9 | 190.31 |
| Hassan | Y = −193.35 + 113.07 *Time + 0.030 *Z450 | Time, Rh1 *Rh2 | 119.76 | 213.9 |
| Kodagu | Y = −69.33 + 16.88 *Time + 6.29 *Z50-0.18 * Z250 + 0.01 *Z341 | Time, Rh2, Tmin *Rh2, Rf *Rh1 | 64.6 | 175.83 |
| Mysuru | Y = −215.62 + 18.56 *Time + 9.71 *Z51 + 5.56 *Z121 | Time, Rh2, Tmax *Tmin | 58.53 | 250.23 |
| Shivamogga | Y = −773.35 + 195.47*Z11 | Tmax | 169.45 | 218.64 |
| Udupi | Y = −162.82 + 49.15 *Time + 5.02 *Z121 + 0.27 *Z141 + 0.007*Z230 | Time, Tmax *Tmin, Tmax *Rh, Tmin *Rf | 217.11 | 94.99 |
| Uttara Kannada | Y = −1285.31 + 16.73 *Z20 + 199.08 *Z21 | Tmin | 42.2 | 244.28 |

The resulting stepwise multiple linear regression model was validated for the period from 2018 to 2019 at the preharvest stage to determine the accuracy of the models. The district-wise predicted rice yield deviation from the observed yield is depicted in Table 5. The yields were underestimated by the model. The error percentages for Kodagu, Mysuru, Udupi, Uttara Kannada, Dakshina Kannada, Ballari, and Belagavi were found to be −34.5%, −2.1%, −1.4%, −6.2%, −1.2%, −8.4%, and −65.1%, respectively, while for rest of the districts it showed overestimations during 2018. Similarly, during 2019 five districts underestimated the rice yield, with error percentages of −1.2%, −2.4%, −8.5%, −13.0%, and −42.3% in Kodagu, Shivamogga, Uttara Kannada, Ballari, and Belagavi, respectively, whereas for the other six districts the predicted yields were overestimated by the model, ranging from 1.3 to 14.9 percent. The results revealed that there seemed to be less agreement between the observed and the predicted yield, as the error calculated by this model was

not found within the acceptable limit, i.e., ±10% for all districts, whereas for a few districts it showed excellent agreement between the observed and predicted yields.

**Table 5.** District-wise error percentage of *Kharif* rice yield (Kg/ha) at preharvest (F3) stage, validated for 2018 and 2019 using a stepwise multiple linear regression.

| District | 2018 | | | 2019 | | |
|---|---|---|---|---|---|---|
| | Predicted Yield (kg/ha) | Observed Yield (kg/ha) | Error (%) | Predicted Yield (kg/ha) | Observed Yield (kg/ha) | Error (%) |
| Ballari | 3926 | 4256 | −8.4 | 4046 | 4571 | −13.0 |
| Belagavi | 1608 | 2655 | −65.1 | 1853 | 2637 | −42.3 |
| Chikkamagaluru | 2937 | 2532 | 13.8 | 2663 | 2532 | 4.9 |
| Dakshina Kannada | 3117 | 3154 | −1.2 | 3155 | 3022 | 4.2 |
| Davanagere | 3815 | 3187 | 16.5 | 3863 | 3514 | 9.0 |
| Hassan | 3252 | 2482 | 23.7 | 3642 | 3101 | 14.9 |
| Kodagu | 2192 | 2948 | −34.5 | 2662 | 2695 | −1.2 |
| Mysuru | 3100 | 3166 | −2.1 | 3521 | 3314 | 5.9 |
| Shivamogga | 2784 | 2669 | 4.1 | 2825 | 2893 | −2.4 |
| Udupi | 3059 | 3101 | −1.4 | 3071 | 3032 | 1.3 |
| Uttara Kannada | 2065 | 2194 | −6.2 | 2214 | 2403 | −8.5 |

The model performance was evaluated with the $R^2$, RMSE, and correlation coefficient (CC). The RMSE ranged between 83.75 and 447.25. Here, a lower RMSE was observed in Udupi district, and a higher RMSE was found in Belagavi district, as an RMSE value close to 0 indicates better model performance. Meanwhile, the CC ranged between 0.51 and 0.95, and the $R^2$ ranged between 0.60 and 0.98. An $R^2$ above 0.6 is said to be a good fit, whereas an $R^2$ between 0.4 and 0.6 is moderate. From the table, we can observe that all eleven districts showed $R^2$ values above 0.6 (Table 6).

**Table 6.** Statistical evaluation of *kharif* rice yield using stepwise regression.

| District | $R^2$ | RMSE | Correlation Coefficient (CC) |
|---|---|---|---|
| Ballari | 0.94 | 188.60 | 0.90 |
| Belagavi | 0.60 | 447.25 | 0.70 |
| Chikkamagaluru | 0.84 | 236.12 | 0.80 |
| Dakshina Kannada | 0.91 | 171.52 | 0.89 |
| Davanagere | 0.95 | 175.44 | 0.70 |
| Hassan | 0.94 | 199.92 | 0.93 |
| Kodagu | 0.91 | 162.28 | 0.80 |
| Mysuru | 0.87 | 235.92 | 0.68 |
| Shivamogga | 0.90 | 212.48 | 0.51 |
| Udupi | 0.98 | 83.75 | 0.95 |
| Uttara Kannada | 0.78 | 234.30 | 0.65 |

3.3.2. Artificial Neural Network Model

A feed-forward neural network with a single hidden layer with eight neurons was fitted with a 'logistic activation function' (default in r package 'nnet') using a calibration/training dataset from 2001 to 2017 (Figure 6). The resulting artificial neural network model was validated/tested with a test dataset for the period from 2018 to 2019 to determine the prediction accuracy of the model. Here, the district-wise predicted rice yield deviated from the observed yield, as depicted in Table 7. During 2018, the negative observed error for yields indicated underestimations by the model in Mysuru (−0.2%), Uttara Kannada (−1.5%), Hassan (−0.1%), Ballari (−1.5%), and Belagavi (−15.3%), while for rest of the districts it showed overestimations by the model ranging from 0.0 to 4.2. Likewise, during 2019, six districts had rice yield underestimations, i.e., Kodagu (−0.6%), Shivamogga (−0.1%), Davanagere (−0.7%), Hassan (−0.2%), Ballari (−5.1%), and Belagavi (−10.8%) districts, whereas for the other five districts the predicted yields were overestimated. Figure 7a,b

reveals the excellent agreement between the observed and predicted yields using SMLR and ANN. But more accurate estimates were found in ANN model. As the error calculated by this model was within the acceptable limit, i.e., ±10% for all the districts and this model can be used to predict yields and was superior to the SMLR.

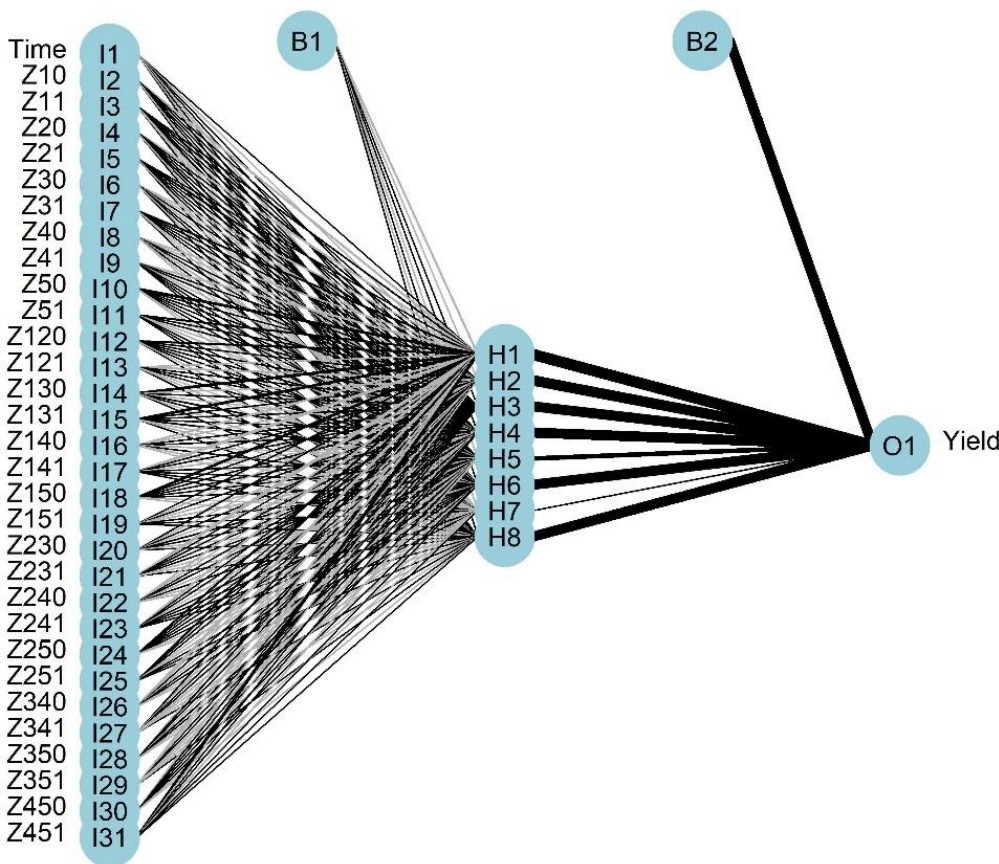

**Figure 6.** Graphical representation of the established neural network for rice yield forecasting Z indicates the weather indices used for the forecast; Ref. Table 2.

**Table 7.** District-wise error percentage of *Kharif* rice yield (Kg/ha) forecasted using artificial neural network (ANN) during 2018 and 2019.

| District | 2018 | | | 2019 | | |
|---|---|---|---|---|---|---|
| | Predicted | Observed | Error (%) | Predicted | Observed | Error (%) |
| Ballari | 4192 | 4256 | −1.5 | 4351 | 4571 | −5.1 |
| Belagavi | 2303 | 2655 | −15.3 | 2379 | 2637 | −10.8 |
| Chikkamagaluru | 2644 | 2532 | 4.2 | 2652 | 2649 | 0.1 |
| Dakshina Kannada | 3153 | 3154 | 0.0 | 3029 | 3022 | 0.2 |
| Davanagere | 3203 | 3187 | 0.5 | 3490 | 3514 | −0.7 |
| Hassan | 2480 | 2482 | −0.1 | 3096 | 3101 | −0.2 |
| Kodagu | 2949 | 2948 | 0.03 | 2678 | 2695 | −0.6 |
| Mysuru | 3159 | 3166 | −0.2 | 3482 | 3314 | 4.8 |
| Shivamogga | 2670 | 2669 | 0.04 | 2889 | 2893 | −0.1 |
| Udupi | 3101 | 3101 | 0.0 | 3031 | 3032 | 0.0 |
| Uttara Kannada | 2161 | 2194 | −1.5 | 2469 | 2403 | 2.7 |

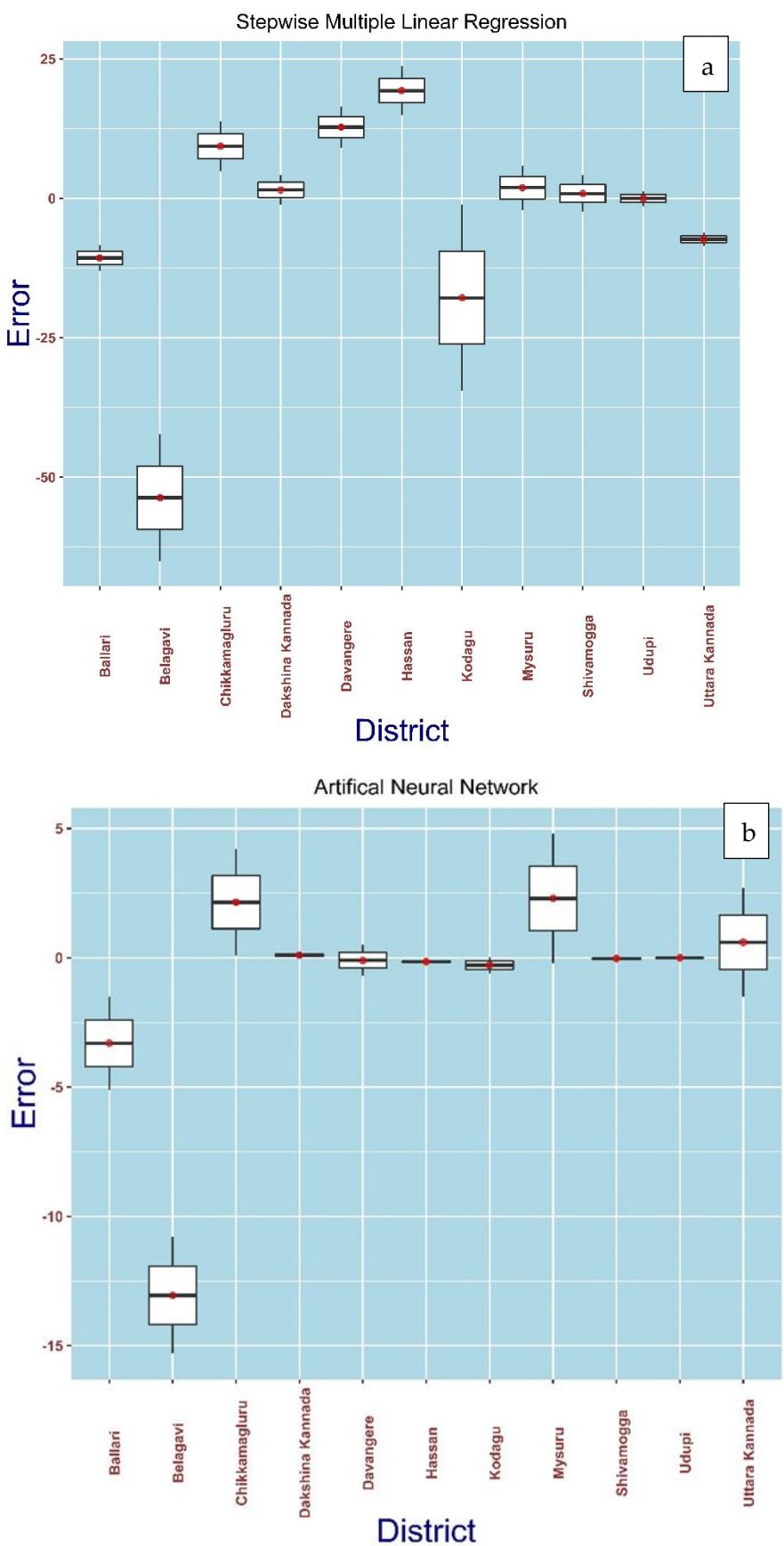

**Figure 7.** Boxplots of district-wise error percentages in *Kharif* rice yields (kg/ha) validated (2018 and 2019) using a stepwise multiple linear regression (**a**) and an artificial neural network (**b**).

The prediction abilities of the fitted ANN models were evaluated in terms of the coefficient of determination ($R^2$), the root-mean-square error (RMSE), and the mean absolute percentage error (MAPE) [43]. Here, the performance analysis included the computation of different statistical parameters, viz. the mean absolute error (MAE), root-mean-square error (RMSE), and normalized root-mean-square error (nRMSE) values for rice crops of different locations (Table 8). A model with smaller RMSE, nRMSE, and MAE values and higher EF values was considered to be the best. The model's performance using ANN as indicated was validated using RMSE values between 1.60 and 281.01; nRMSE values between 0.06 and 15.40; MAE values between 1.22 and 187.72, and EF values between 0.80 and 1.00. Among the predicted districts yields, the lowest values of RMSE (1.60), NRMSE (0.06), MAE (1.22), and EF (1.00) were found in Shivamogga district, and the highest were observed in Belagavi district with 281.01, 15.40, 187.72, and 0.80 for RMSE, NRMSE, MAE, and EF, respectively. The model was said to perform excellently, with an nRMSE value less than 10 percent, categorized as excellent, for ten out of eleven districts and an nRMSE value categorized as good for one district, as it was between 10 and 20 percent. This observed variability among the models was due to the consideration of less independent parameters by SMLR compared to ANN, which took into account multiple interactions between the weather variables [44].

**Table 8.** Statistical evaluation of validated *kharif* rice yield using an artificial neural network (ANN).

| District | RMSE | nRMSE | nRMSE * | MAE | EF |
|---|---|---|---|---|---|
| Ballari | 88.61 | 2.70 | Excellent | 46.04 | 0.96 |
| Belagavi | 281.01 | 15.40 | Good | 187.72 | 0.80 |
| Chikkamagaluru | 99.75 | 3.91 | Excellent | 53.23 | 0.94 |
| Dakshina Kannada | 15.21 | 0.61 | Excellent | 6.93 | 1.00 |
| Davanagere | 28.34 | 0.80 | Excellent | 20.79 | 0.99 |
| Hassan | 6.88 | 0.25 | Excellent | 4.66 | 1.00 |
| Kodagu | 71.99 | 2.72 | Excellent | 42.86 | 0.93 |
| Mysuru | 44.45 | 1.40 | Excellent | 29.95 | 0.98 |
| Shivamogga | 1.60 | 0.06 | Excellent | 1.22 | 1.00 |
| Udupi | 3.83 | 0.15 | Excellent | 1.99 | 1.00 |
| Uttara Kannada | 131.06 | 6.45 | Excellent | 72.77 | 0.82 |

* nRMSE classes: <10% = Excellent, 10–20% = Good, 20–30% = Fair, and >30% = Poor.

### 3.4. Effect of Weather Variables on Rice Yield

Rice, being one of the major food grain crops whose productivity is largely dependent on weather variables, has shown yield variability due to the impacts of individual and interactive effects of different weather variables. The average weekly temperature (Tmean) of the study regions during the rice growing period varied between 22 and 26 °C (Table 9), which is very much within the optimal temperature range required for rice growth from 15–18 to 30–33 °C, [45]. The average weekly maximum temperature (Tmax) ranged between 26 and 30 °C, and previous studies have observed yield variability of rice due to temperature fluctuations [46–48]. Sometimes the maximum temperature exceeded 35 °C, which has caused destructive effects on rice growth and yield [49]. This may be due to alterations in enzyme activities, leading to changes in the rate of photosynthesis, respiration, and other physiological aspects [17,50]. Higher temperatures were found to decrease the duration of the crop life cycle, thereby shortening the grain filling period, which might lead to lower crop yields and grain quality. The RHI ranged between 69 and 94%, and the RHII ranged between 59 and 91%. Higher relative humidity has an antagonistic effect on crop yield, as higher humidity causes a reduction in evapotranspiration, thereby lowering the cooling effect due to evaporation [51]. It also supports incidences of pests and diseases, which lead to reductions in crop yield. However, a higher vapor pressure deficit during anthesis leads to a reduction in the panicle temperature due to transpirational cooling, which helps in reducing high-temperature-induced spikelet sterility [52]. The average rainfall in the region varied between 27.7 and 181.8 mm during the crop growth period

(Table 9). In districts such as Ballari, the rainfall was much too low to support crop growth, even though the yield levels were high. This was merely due to the availability of sufficient irrigation water through reservoirs. This strengthens crop management, as rainfall during the flowering and ripening stages reduces pollination and causes lodging [53].

**Table 9.** Statistics of weather variables observed during the crop growth period.

| Statistic | Tmax (°C) | Tmin (°C) | Tmean (°C) | RHI (%) | RHII (%) | Rainfall (mm) |
|---|---|---|---|---|---|---|
| Mean | 27.9 | 20.8 | 24.3 | 87.1 | 79.0 | 85.6 |
| Maximum | 30.6 | 22.5 | 26.4 | 94.0 | 91.5 | 181.8 |
| Minimum | 26.3 | 19.4 | 22.9 | 69.0 | 59.0 | 27.7 |
| Standard deviation | 1.2 | 1.0 | 1.1 | 6.7 | 9.1 | 50.0 |
| Coefficient of variation (%) | 4.4 | 5.0 | 4.5 | 7.7 | 11.5 | 58.4 |

*3.5. Comparison of SMLR and ANN for the Predictability of Regional Rice Yield*

The yields forecasted for rice during 2021 at the preharvest (F3) stage using a stepwise multiple linear regression (SMLR) and an artificial neural network (ANN) for eleven growing districts of Karnataka during *kharif* season are presented in Table 10. The yields forecasted by SMLR ranged from 2294 to 4093 kg/ha. The higher yield was predicted for Udupi (4093 kg/ha) district, followed by Hassan (3929 kg/ha), whereas the lower yield was predicted in Belagavi (2294 kg/ha), followed by Uttara Kannada (2314 kg/ha) district. Further, the ANN forecasted yields ranging from 2172 to 3919 kg/ha, and the higher yield was predicted in Ballari (3919 kg/ha) district, followed by Shivamogga (3586 kg/ha), whereas the lower yield was predicted in Uttara Kannada (3586 kg/ha), followed by Mysuru (2318 kg/ha) district. Meanwhile, the percentage deviations from the observed yield ranged between −2.5 and 35.0% (mean: 16%) in SMLR, and −37.1 to 21.3 percent (mean: 4%) deviations were observed using ANN. These outcomes prove the usability of ANN over SMLR in crop yield forecasting.

**Table 10.** District-wise average *Kharif* rice yield and predicted yields using a stepwise multiple linear regression and an artificial neural network (ANN) during 2021 at the preharvest (F3) stage.

| District | Average Yield * | SMLR | | ANN | |
|---|---|---|---|---|---|
| | | Predicted Yield 2021 (kg/ha) | Deviation % | Predicted Yield 2021 (kg/ha) | Deviation % |
| Ballari | 3302 | 3223 | −2.5 | 3919 | 15.7 |
| Belagavi | 1847 | 2294 | 19.5 | 2348 | 21.3 |
| Chikkamagaluru | 2555 | 2747 | 7.0 | 2447 | −4.4 |
| Dakshina Kannada | 2492 | 3223 | 22.7 | 2753 | 9.5 |
| Davanagere | 3551 | 3834 | 7.4 | 2882 | −23.2 |
| Hassan | 2746 | 3929 | 30.1 | 3166 | 13.3 |
| Kodagu | 2648 | 2591 | −2.2 | 2785 | 4.9 |
| Mysuru | 3177 | 3579 | 11.2 | 2318 | −37.1 |
| Shivamogga | 2872 | 3225 | 10.9 | 3586 | 19.9 |
| Udupi | 2660 | 4093 | 35.0 | 2785 | 4.5 |
| Uttara Kannada | 2044 | 2314 | 11.7 | 2172 | 5.9 |
| Average | 2718 | 3235 | 16.0 | 2833 | 4.1 |

* Observed yield (kg/ha) averaged from 1980 to 2019.

The average district yield forecasted using stepwise multiple linear regression was found to be 3235 kg ha$^{-1}$, and using artificial neural network (ANN) this value was found to be 2833 kg ha$^{-1}$ at the preharvest stage compared to the average yield (1980–2019) of 2718 kg ha$^{-1}$, as depicted in Figure 8. The forecasted mean yield during 2021 using both the methods was found to be higher than the average yield. As the model is purely

weather-based, good rains during the crop growing season could be the reason for higher yield estimates for 2021 in all forecasted districts.

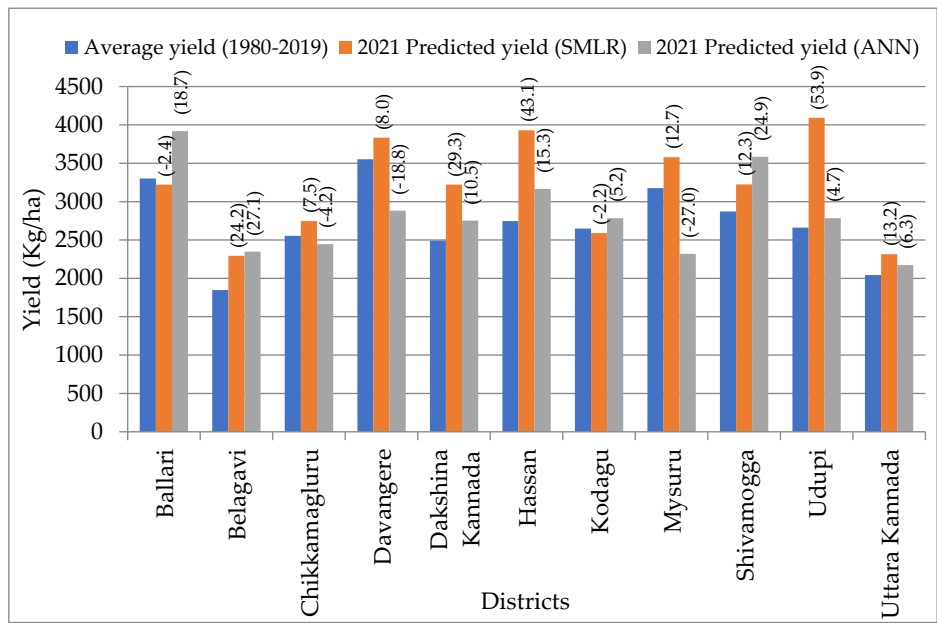

**Figure 8.** District-wise yield prediction for *Kharif* rice (kg/ha) using an artificial neural network and a stepwise multiple linear regression during 2021, along with the average yield comparison. Numbers in parenthesis indicate the deviation percentages from the average yields.

## 4. Discussion

Agriculture is a production sector that is highly dependent on the climatic conditions [54–56]. Especially in tropical countries such as India, where the majority of the crop production is dependent on the climate. While agricultural output is dependent on other factors, such as pest and diseases, weeds and their management decisions, etc., those can be modified in an effort to provide the best growing environment for the crops [57]. Even after providing the best growing environment for the crop, variations in the crop yield are observed. Such variations are majorly attributed to spatial and temporal variations in weather factors [58]. This weather-induced production variability impacts regional food security, thus making it necessary to study the major weather factors behind crop production. The quantification of weather impacts on the crop growth is a cumbersome task, as weather factors impart yields through their direct and interactive effects. The present study involved the use of statistical (SMLR) [59,60] and machine learning tools (ANN) [61] to generate a better rice yield forecast model. As the crop is a staple food crop of the majority of the population in Karnataka, its interaction with weather needs to be assessed to have an advance estimate of its production in the region and to plan the alternatives for improving the productivity of the secured food supply. Previous studies have shown the crop's dependence on weather, but few number studies have been conducted on quantifying the interactive effects of weather on rice crops. Therefore, efforts are being made to assess the interactive effects of weather factors on rice productivity through the generation of weather indices based on composite weather variable methods [62,63] for understanding the joint effect of two variables [64–66].

The SMLR and ANN models were calibrated (1980–2017) and validated (2018–2019) using the historical datasets of weather variables (IMD) and crop yield datasets (state agriculture department) to forecast the 2021 yield. In order to fit the SMLR, the normality in the yield data was checked using the Shapiro–Wilk test and Q-Q plots, which indicated that all districts' yield datasets were normally distributed ($p$ value $> 0.05$ and quantiles centered around the diagonal line) except Mysuru ($p$ value $= 0.001$). The comparison of

the SMLR and ANN models revealed a higher percentage deviation from the observed yield, ranging between $-2.5$ and $35.0$ (mean: 16%), in SMLR compared to ANN (range: $-37.1$ to $21.3$, mean: 4%), making ANN a better model for forecasting. This might be due to the ability of ANN to take an account of the collinearity between weather variables for yield prediction [67,68] Further, the district-wise yield predictability of ANN was assessed using the mean absolute error (MAE), which ranged between 1.22 and 187.72 with a mean of 42. Among the districts, the lowest value of MAE (1.22) was observed in Shivamogga, and the highest was in Belagavi district, indicating better predictability of ANN in these districts, and the mean RMSE obtained using the ANN model was found to be 70 compared to 213 in the regression model [69]. Hence, the use of machine learning tools such as ANN, LASSO, NNET, etc., paves a path for precision yield forecasting, which could be promising for decision making in future crop management and the planning of policies for improved yield production.

## 5. Conclusions

Machine learning approaches are promising alternatives or complimentary tools to support the commonly used crop simulation model for yield prediction, but their efficacy has to be evaluated before applying them to a specific crop or cropping system yield prediction. As a crop's performance is influenced by more than one external factor such as the weather and the interactions between the weather factors themselves, a special method is needed to assess performance. Previously, several linear models were developed based on the direct relationships between the yield and the weather. They failed to measure the impact of multicollinearity between weather factors on the yield. Therefore, an attempt to determine this impact using a machine learning tool (ANN) and compare it with a simple regression model such as SMLR was made in the present study. Two multivariate models, SMLR and ANN, were used to forecast rice yields in major rice-growing districts of Karntaka. The outcomes demonstrated that an artificial neural network (ANN) can be utilized for yield prediction for the area with satisfactory results compared to a stepwise multiple linear regression since a good agreement was realized between the ANN-predicted and the observed yield, which was indicated by the root-mean-square error, normalized root-mean-square error, $R^2$ statistics, and prediction error percentage. There are various advantages of ANNs over conventional approaches; they provide a stable analytical alternative to conventional regression techniques, which are often limited by strict assumptions of normality, linearity, variable independence, etc. As ANNs are able to capture interactions between independent variables, they allow a quick and easy method for modeling the complex agricultural phenomenon that is otherwise nearly impossible to explain.

**Supplementary Materials:** The following are available online at https://www.mdpi.com/article/10.3390/agronomy13030704/s1, Table S1: Long-term yield and weather dataset used for the analysis.

**Author Contributions:** M.N.T. and M.H.M. resources, conceptualization, and validation; L.H. and D.V.S., analysis, investigation, and original draft preparation; H.S.P. and L.N., data curation and original draft preparation; H.S.S., review and editing, visualization, and supervision. All authors have read and agreed to the published version of the manuscript.

**Funding:** This research was not funded by any externally funded project.

**Data Availability Statement:** The data presented in this study are available on request from the corresponding authors.

**Acknowledgments:** The authors are grateful to acknowledge the FASAL-India Meteorological Department and the Directorate of Economics and Statistics for providing the weather and yield data.

**Conflicts of Interest:** The authors declare no conflict of interest.

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
