# Peer review of "Weather-Based Statistical and Neural Network Tools for Forecasting Rice Yields in Major Growing Districts of Karnataka"

_agronomy, doi:10.3390/agronomy13030704_

Round 1
Reviewer 1 Report
The author studied two models (SMLR and ANN models) to forecast the yield of the main crop 'kharif rice' planted in Karnataka, India, based on weather. As mentioned in this paper, weather variables play a vital role in the growth process of crops. As an agricultural and populous country, it is very necessary to study the impact of multiple factors on crop yield in India, so as to regulate agricultural production and achieve the situation of less investment and more production. Therefore, the research in this paper is meaningful. From the results of the paper, the ANN model has a better performance, which is not only conducive to the prediction of local rice yield, but also provides a reference for other regions in model selection. The approach employed and findings results are good and can be published in the agronomy journal.
However, there are still some places in this paper that need to be revised. My suggestions are as follows:
1. Abstract: The abstract of this paper does not mention a clear research scope. The introduction of the methods and conclusions in the abstract part suggests that they should be clearly distinguished, and at the end, the significance of the research results can be explained in a simple sentence.
2. Add ‘weather ’to keywords.
3. Introduction: The contents in lines 72-83 are suggested to be described at the beginning of the introduction section as a research background. The introduction of SMLR and ANN in lines 60-70 should be more detailed and clearer.
4. The district names in Table 1 should correspond to those in Figure 1. 'Shimoga ' and ' shivamogga ', ' chikkamagalur ' and ' chikkamagaluru ' refer to the same district?
5. In Figure 2, 'daliytmax, daliytmin, daliyrf, daliyrhi, daliyrhii' are defined above? Improve the performance of Figure 2.
6. Adjust the relative size of each component in Fig. 3. The bar graph on the right is too small and the label below is too large, which will further improve the whole graph. Figures 4, 5 and 6 also have the same problem. It is recommended to improve the performance of the figures as a whole.
7. Tables 2 to 10 have the same problem as table 1, and check the unit of data in the table.
8. Can 2.3.2. (Simple Multiple Linear Regression (SMLR)) and 2.3.3. (Checking the normality of yield data) be described together?
9. 2.3.4.(Artificial Neural Networks (ANN)) when introducing ANN model, it is suggested to add a graph to show it, and specifically explain the division of data set. Which parts are used for training and which are used for testing?
10. Line 213-216: The authors need to provide brief explanation equations and why they use three statistical measures (R2; RMSE; nRMSE; and EF) for the reader better understanding.
11. The discussion section, lines 474 to 477, can specifically recommend several machine learning models that may have better performance.
12. In the conclusion part, it is suggested to add a short introduction to how to make readers use the research results of this paper, as well as such meaningful and beneficial aspects.
Reviewer 2 Report
Brief summary
The paper compares linear models and artificially neural models to predict rice yield in Southern India. It seems that, at the best of my knowledge on the topic, the combine use of official statistics and weighted weather indices are the main contribution of this paper. In general, the concepts expressed in the paper are current and well suited to be published in Agronomy.
General concept comments
I found difficult to understand what type of artificial neural network (ANN) was used. From the manuscript is not clear if the ANN is shallow or deep, what type of layers have been chosen, how many neurons are in each layer, what are the activation functions considered... All these missing details do not allow one to implement the same ANN from scratch.
While ANN have been known to outperform linear models when data relationships are nonlinear, it seems that different types of networks have not been fully explored. E.g., if time is consider (as shown in Table 4), LSTM networks have been found to outperform standard ANN.
Furthermore, there is no discussion on the chosen training algorithm for the network model, and what settings was used. ANN (and other types of neural networks) are also well-known to overfit the training set. However, the usual graphics showing the history of the loss (used for training) and the metrics (used for validations) are not shown.
Specific comments
- I am not convinced that all reference are appropriate. E.g., some citations may not peer-reviewed, others are used for general statements but they are too specific to a narrow application (such as [2]).
- (Table 1) I do not understand the notation (000 ha). Do the authors mean (1000 ha)? The same appears on the unit of measure for production.
- (Table 2) The weather "parameters" (should be called variables throughout the whole manuscript) have acronyms that should be introduced in the main text. E.g., RF, RHI and RHII are not that clear when compared to the text of line 110 and 111.
- (Section 2.3.3, lines 263-268 and lines 460-463) Linear models are usually fitted using the ordinary least square (OLS) estimator under Gaussian assumptions; however, the OLS can be considered as a distribution-free approach that minimize the MSE independently of the underlying data distribution. This means that a linear model fitted with the OLS produces an approximation of the conditional expectation of the response for a give set of input variables. However, if the authors consider important to draw conclusions on the regression coefficients based on standard statistical literature, a different normality test (e.g., Shapiro test) would a better choice. Otherwise, the normality checks are superfluous and should be removed.
- (Line 244) More references are needed.
- (Lines 323-324) The description of the ANN model is not sufficient. It needs to be extended (please, see my general comments).
- Figure 5 should contain brief titles on top of the two box-plot graphics.
- (Figure 6) please, consider to replace "UK" with "Uttara K." and "DK" with "Dakshina K." and to move the legend above the top line at 4500 kg/ha.
Reviewer 3 Report
The presented article used two multivariate models SMLR and ANN for rice yield forecast. The article is interesting, however, some issues need to be addressed by the Authors:
Page 1, lines 38-41, provide references for every multidisciplinary source of information stated in the sentence.
Page 2, lines 70-72, the end of the previous passus and the beginning of this one is disconnected, try to add a brief transitional sentence connecting the selection of crops and statistical models.
Page 2, line 72, add a scientific name for rice
Page 3, Figure 1, mark percent rice area and percent contribution to state’s rice production with a and b and use the same font and letter size for each figure
Page 4 and 6, formulas should be standardized and marked according to the requirement of the journal (refer to the Guide of Authors)
Page 5, 172, Figure 2, and Table 2 should appear below the text where were first mentioned
Page 9, discus high CV values from Table 3 for Ballari and Belgavi districts
Page 12, provide artificial neural network model summary (performance and errors),
Page 15, add statistical deviation on figure 6
Round 2
Reviewer 1 Report
The reversion from the authors has been received, and the authors have made a lot of modifications, which has improved the overall paper. However, there are still some imperfections in the modification, which can be further improved. The suggestions are as follows:
1. The most critical point is the lack of novelty in the work. The models in the analysis are normal and the potential contribution of the results is not clear, which mitigated against publication in Agronomy.
2. The structure of the draft and the figures also need to be revised. The Figures can hardly meet the standard of scientific publishing. All Figures should be revised to higher resolution. Poor resolution hard to read.
3. Introduction: Introduction should be rewritten. The purpose of the work, its significance and novelty should be highlighted, especially the novelty with the previous studies using neural networks.
.
4. Figure 1. Per cent rice area (left) and per cent contribution to state’s rice production (right) by each district of study area (Source: Rice area and production (2021), Directorate of Economics and Statistics, GoK). We can use a and b instead of left and right.
5. 2.3.3. Artificial Neural Networks (ANN): This part should add a diagram (Artificial neural network architecture example) to better show the establishment of the neural network in this paper.
6. Figure 3 is still not very good. It is suggested to consider the overall expression of Figure 3 and change the expression mode.
7. Figure 4: Poor resolution hard to read.
8. The modification of Figure 5 is not only to enlarge it, but also to improve the rendering effect of the figure from the coordinate axis scale, labels and background.
9. There are still problems with the changes in Figure 6. Such changes are invalid.
10. The author needs to support the discussion with reference paper, which will give high value to the manuscript. I recommend to add 2-5 references.
Ge, J., Zhao, L., Gong, X., Lai, Z., Traore, S., Li, Y., Long, H. and Zhang, L., 2021. Combined effects of ventilation and irrigation on temperature, humidity, tomato yield, and quality in the greenhouse. HortScience, 56(9), pp.1080-1088.
Traore, S., Zhang, L., Guven, A. and Fipps, G., 2020. Rice yield response forecasting tool (YIELDCAST) for supporting climate change adaptation decision in Sahel. Agricultural Water Management, 239, p.106242.
Reviewer 2 Report
Brief summary
The paper compares linear models and artificially neural models to predict rice yield in Southern India. It seems that, at the best of my knowledge on the topic, the combine use of official statistics and weighted weather indices are the main contribution of this paper. In general, the concepts expressed in the paper are current and well suited to be published in Agronomy.
General concept comments
I found a bit disappointing seen that the neural networks have not fully explored using LSTM layers or other type of recurrent layers. However, I found the paper to be improved in many of the other aspects that I mentioned in my previous review.
Although, I still think that the analysis with the neural network models is not expressing its full predictive potential due to the lack of consideration for more state-of-the art neural models, I reconsidered my previous decision to reject this paper for publication. In its current form, I would suggest minor revisions before publication.
Please, consider to address the following "specific comments".
Specific comments
(Line 99) "overdependence on independent factors" is confusing. First, multicollinearity among the variables in input does not cause over-fitting. Second, if the factors are independent, how can they correlated to each other? Please, consider rephrasing this sentence... possibly by referring to the aspect of having more variables than data points (which seems to be a more likely scenario where feature selection is required).
(Line 101-104) ANN are not used for feature selection. Furthermore, I do not understand how the feature selection methods "have been evolved...", maybe "have been adapted to increase the precision of yield forecast"?
(Line 428) "a signle layer" with... how many neurons? Line 431 states that the ANN have been validated and tested, but shallow networks are usually validated to chose the number of neurons in the hidden layer.
Reviewer 3 Report
The author's responses are fair. The article can be accepted in its present form.
Round 3
Reviewer 1 Report
I think the paper was well revised. However, my concern is the lack of novelty in the work. Significance and novelty should be highlighted in the introduction, especially the novelty with the previous studies using neural networks. Thanks
Author Response
Response:
Sir,
- Thanks for accepting the revisions and for appreciating.
- Regarding the introduction part it has been modified as per your instructions by highlighting the previous studies done on Artificial neural network.
